# S-Acylation of Proteins of Coronavirus and Influenza Virus: Conservation of Acylation Sites in Animal Viruses and DHHC Acyltransferases in Their Animal Reservoirs

**DOI:** 10.3390/pathogens10060669

**Published:** 2021-05-29

**Authors:** Dina A. Abdulrahman, Xiaorong Meng, Michael Veit

**Affiliations:** 1Department of Virology, Animal Health Research Institute (AHRI), Giza 12618, Egypt; dinaaly89@gmail.com; 2Institute of Virology, Veterinary Faculty, Free University Berlin, 14163 Berlin, Germany; xiaorong.meng@fu-berlin.de

**Keywords:** coronavirus, influenza virus, S, E, HA, M2, S-acylation, palmitoylation, DHHC, pandemic preparedness

## Abstract

Recent pandemics of zoonotic origin were caused by members of coronavirus (CoV) and influenza A (Flu A) viruses. Their glycoproteins (S in CoV, HA in Flu A) and ion channels (E in CoV, M2 in Flu A) are S-acylated. We show that viruses of all genera and from all hosts contain clusters of acylated cysteines in HA, S and E, consistent with the essential function of the modification. In contrast, some Flu viruses lost the acylated cysteine in M2 during evolution, suggesting that it does not affect viral fitness. Members of the DHHC family catalyze palmitoylation. Twenty-three DHHCs exist in humans, but the number varies between vertebrates. SARS-CoV-2 and Flu A proteins are acylated by an overlapping set of DHHCs in human cells. We show that these DHHC genes also exist in other virus hosts. Localization of amino acid substitutions in the 3D structure of DHHCs provided no evidence that their activity or substrate specificity is disturbed. We speculate that newly emerged CoVs or Flu viruses also depend on S-acylation for replication and will use the human DHHCs for that purpose. This feature makes these DHHCs attractive targets for pan-antiviral drugs.

## 1. Introduction

### 1.1. Pandemics Caused by Influenza Virus and Coronaviruses

A pandemic is caused by the emergence of a new and highly transmissible virus in immunologically naïve humans. In the absence of antivirals, vaccines or non-pharmaceutical interventions, such a virus will infect a large proportion of the population until increasing “herd immunity” will slow down further virus spread. Influenza A viruses caused four pandemics in the last 100 years, most notably the deadly “Spanish Flu” from 1918/19 [1,2,3]. Pandemic viruses originated from water birds of the orders *Anseriformes* (ducks) and *Charadriiformes* (shorebirds, gulls). Occasionally, influenza A viruses leave their animal reservoirs, and jump to either poultry, where they may cause an outbreak of deadly “bird-flu”, or to various mammalian species, especially humans and pigs. Pigs are susceptible to both avian and human viruses without prior adaption and, hence, are considered as a “mixing vessel” for the generation of pandemic Flu viruses [4]. In addition, exchange of gene segments between both viruses (“antigenic shift”) creates a reassortant, that might be better adapted to infect humans [5,6,7]. Recently, viruses belonging to two different lineages of Flu A were also identified in flat-faced fruit bats (*Artibeus planirostris*) and littel yellow-shouldered bats (*Sturnira lilium)*; both belong to the family Phyllostomidae, which are abundant throughout Central and South America [8,9]. Since the bat-associated viruses use different receptors than the other influenza viruses (MHC class II versus sialic acid-containing receptors) [10], their potential for human emergence is currently under investigation. 

*Coronaviruses* (CoVs), which are divided into the genera α, β, γ and δ-CoVs, include several pathogens of mammals (α- and β-CoVs) and birds (γ- and δ-CoVs), where they cause gastrointestinal or respiratory infections with a wide range of clinical manifestations. Their relevance for human health, however, was limited until 2002, when the outbreak of SARS occurred in Southern China. The causal agent, *SARS-CoV-1*, a β-CoV, originated from *Rhinolophus* bats (family *Rhinolophidae*, suborder *Yinpterochiroptera*), its animal reservoir [11], but was introduced into humans via amplifying hosts, palm civets and raccoon dogs [12], where it acquired adaptive mutations. After infection of ~8000 patients in 29 countries, the virus was finally eradicated by public health measures [13]. However, *Rhinolophus* bats in Southern China harbor many SARS-related viruses—some are already able to infect human cells [14]. The frequent exchange of genetic material between viruses by recombination increases the viral gene pool further. 

Another β-coronavirus, MERS-CoV was first isolated in 2012 in Saudi Arabia from a human patient [15], who was infected by contact with dromedary camels, its main reservoir [16]. The virus most likely had its origin in *Vespertilionidae* bats (family Vespertilionidae, suborder *Yangochiroptera*), especially in the species *Tylonycteris pachypus*, *Pipistrellus abramus* and *Neoromicia zuluensis* [17,18,19], before it was transmitted to camels. Multiple transmissions of MERS to humans occurred, mostly in the Arabian Peninsula, but only human-to-human transmissions were recorded. However, the frequency of sporadic outbreaks indicates that MERS remains a significant threat, especially if the virus becomes more transmissible [20].

Finally, the highly transmissible SARS-CoV-2 appeared in December 2019 in Wuhan and caused the first coronavirus pandemic [21]. The closest relative of SARS-CoV-2 is again a virus from *Rhinolophus* bats [22,23], but similar viruses identified in pangolins might also play a role in its emergence [24]. There might be the risk for reverse zoonotic transmission of SARS-CoV-2 to bats in North America, where they might establish a new wildlife reservoir [25]. Furthermore, the four “common cold” now-endemic human coronaviruses, hCoV-229E, -NL63, -OC43, and -HKU1, might have jumped from livestock species to humans, but two of them (hCoV-NL63 and -229E) had their original animal reservoir in bats [26]. 

### 1.2. S-Acylation of Influenza Virus and Coronavirus Proteins

In order to jump frequently between species, viruses must be able to replicate in cells from evolutionarily distant organisms. This implies not only that these cells must express a suitable virus receptor, but must also perform all the essential modifications of viral proteins with high efficiency. Such a modification is the attachment of fatty acids to the main glycoproteins of influenza and coronaviruses, the hemagglutinin (HA) and spike (S), respectively, and to their ion channels M2 and E [27]. Both HA and S are trimeric type 1 transmembrane glycoproteins with an N-terminal signal peptide, a large ectodomain, a single TMR (transmembrane region) and a short cytoplasmic tail. They are cleaved by cellular proteases into the HA1/S1 subunit, which contains the receptor binding site and the antigenic epitopes, and the membrane-anchored HA2/S2 subunit, which harbors the membrane fusion machinery [28,29,30]. M2 and E are composed of a short hydrophilic N-terminal region, one TMR and a longer cytoplasmic tail. Both proteins are oligomers, their TMRs form an ion-channel that is involved in virus entry. Their cytoplasmic tails bind to other viral proteins and hence organize virus assembly and contain a membrane-near amphiphilic helix which is involved in virus budding and release [31,32]. 

HA of human influenza A viruses is S-acylated at two cytoplasmic cysteines with palmitate (C 16:0) and at one transmembrane cysteine primarily with stearate (C18:0). HA of influenza B virus, which circulates only in humans where it contributes to the seasonal Flu, contains only the two palmitoylated cytoplasmic cysteines and the hemagglutinating-esterase-fusion glycoprotein HEF of influenza C, which is also a human virus, but causes only mild symptoms [33], is predominantly stearoylated at one cysteine at the end of the TMR [34,35]. The S proteins of corona viruses are acylated at a cluster (7–10) of cysteines located the beginning of the cytoplasmic tail, but the attached fatty acid species have not been determined [36]. Likewise, M2 and E are acylated at one and up to three cysteines, respectively, located adjacent to the TMR at the beginning of an amphiphilic helix [36,37,38] (Table 1).

Work from several groups have shown that S-acylation of HA of Flu A is essential for virus replication. Two human viruses, one of the subtype H1 and the other of the subtype H3, as well as one avian virus of the subtype H7, could not be generated from plasmids if two or three acylation sites were removed from HA. In addition, mutants with one deleted acylation site revealed greatly reduced growth rates in cell culture [39,40,41,42]. Likewise, similar genetic studies performed with a prototype coronavirus (murine hepatitis virus, MHV) exhibits greatly reduced virus titers and infectivity if individual acylation sites in S were exchanged in the viral genome [43]. Functional studies in transfected cells with S of both SARS-CoV-1 and SARS-CoV-2 (and other animal CoVs) revealed that palmitoylation affects its core activity, membrane fusion, and also binding to the M protein and, hence, virus assembly [44,45,46,47,48,49,50]. Similar results were obtained for HA of Flu A, suggesting that the molecular mechanism how palmitoylation might affect the function of both proteins might be very similar. Palmitoylation of E also has a strong effect on virus replication. Virus titers were greatly reduced upon exchange of two or all three acylation sites in the genome of MHV and the expressed protein did not recruit the M protein indicating that palmitoylation affects virus assembly [44,51]. In contrast, virus mutants with deleted acylation site in M2 revealed small growth defects in cell culture, but were slightly attenuated in mice [52,53,54]. 

### 1.3. Structure and Function of DHHC Proteins 

Palmitoylation of cellular proteins is catalyzed by members of the family of DHHC proteins, but the ones involved in acylation of viral proteins have remained elusive until recently. DHHC proteins are polytopic membrane proteins containing an Asp-His-His-Cys (DHHC) motif as the catalytic center in one of their cytoplasmic domains, which is embedded within a ~50 amino-acid-long, cysteine-rich domain (CRD). Twenty-three DHHC proteins exist in humans (DHHC1 to DHHC24—DHHC10 does not exist) which do not exhibit much sequence conservation besides the cysteine-rich domain. Most DHHCs are abundant in many tissues, where they mainly localize to Golgi membranes. A smaller number remain in the endoplasmic reticulum (ER) or a targeted to the plasma membrane or to endosomes. Most cellular proteins can be palmitoylated by several, but not each of the various DHHC proteins. This indicates that the 23 enzymes show distinct, only partially overlapping substrate specificities [55,56,57,58]. 

The crystal structures of two DHHC proteins, zebrafish DHHC15 and human DHHC20 [59] revealed that four transmembrane helices form a tent-like structure with the DHHC-motif (residues 153–156, shown as blue sticks in the cartoon model of DHHC20, Figure 1) located at the membrane–cytosol interface. The CRD (shown in light green) is positioned between TMR 2 and 3. It contains three α-helices and then forms six β-sheets that coordinate two zinc ions via highly conserved CCHC zinc-finger domains (residues shown as yellow sticks). The active site cysteine does not coordinate any of the ions indicating that zinc does not directly participate in catalysis. The CRD also contains a patch of positively charged residues (Arg 126, Lys 135, His 140, shown as orange sticks) that are thought to bind the negatively charged phosphates of the acyl-CoA moiety, the lipid donor. Many DHHCs exhibit a two-step reaction mechanism, a fatty acid is first transferred from acyl-CoA to the cysteine of the DHHC motif (auto-acylation) and subsequently to the substrate protein [60]. The acylated enzyme intermediate contains the fatty acid (labelled in red) inserted into a cavity formed by hydrophobic amino acid side chains of all four transmembrane regions. The narrow end of this tunnel of DHHC20 is sealed by a hydrogen bond between Ser 28 and Tyr 181 (labelled as blue sticks). In contrast to many other amino acids contacting the acyl chain, Ser 29 and Tyr 181 are not conserved between DHHC proteins; all DHHCs contain either two bulky residues, one bulky and one small or two small amino acids at the homologues position. It was hypothesized that the presence of certain amino acids (large or small) determines the lipid binding specificity of a DHHC, i.e., whether palmitate (C16) or stearate (C18) are preferentially attached [61]. The C-terminal domain is more variable between DHHC enzymes, but contains certain sequence motifs conserved to varying extents throughout the DHHC family. Immediately adjacent to TMR 4 is a conserved TTXE motif (labelled in cyan). Thr 141 forms a hydrogen bond with Asp and His of the catalytic center and is, thus, likely to be involved in the catalytic reaction. The C-terminal domain continues with an amphipathic helix and a hydrophobic loop that together form a supporting structure for the transmembrane domain (shown in purple). The last ~30 residues, predicted to be unstructured, are missing from the hDHHC20 structure. Besides this core domain, some DHHCs contain N- or C-terminal extensions that carry ankyrin, SH-2 or PDZ domains that are presumably required to recruit cytosolic proteins to the enzyme to facilitate acylation [62,63]. 

During evolution, DHHCs appeared first in eukaryotes. Yeast encode five to seven DHHC members, plasmodium parasites twelve [64] and metazoan genomes have up to 30 DHHC genes. However, the number of DHHCs does not gradually increase during evolution. Instead, a bioinformatic study showed that duplications and losses of DHHC genes occurred and thus there is considerable variation between species [65].

### 1.4. DHHCs Involved in Acylation of Influenza and Coronavirus Proteins

The DHHCs that catalyze acylation of viral proteins are now beginning to be elucidated. Using siRNA screens and CRISPR/Cas9 knockout revealed that DHHC2, 8, 15 and 20 are involved in acylation of HA and M2 of an avian and also of a human Influenza virus, individual DHHCs exhibit only slightly different substrate preferences. However, none of these enzymes is required for acylation of HA of Flu B and HEF of Flu C, demonstrating that HA of Flu A must contain a certain recognition element which is absent in HA of Flu B and in HEF [66]. Using similar technology, it was recently reported that S of SARS-CoV-2 is mainly acylated by DHHC20, but DHHC8 and 9 also play a role. Likewise, E of SARS-CoV-2 is acylated by a similar set of DHHC enzymes [50].

### 1.5. Rationale for This Study

We have recently proposed that DHHCs required for acylation of HA could be promising drug targets, since their blockade should result in suppression of viral replication, while acylation of cellular proteins will not be (or very slightly) compromised [67,68]. Since SARS-CoV-2 and Influenza viruses are acylated by an overlapping set of DHHCs, one can imagine that the same drug might work against both viruses. Furthermore, such a compound might even be effective against newly emerged viruses that jump from animals to humans. The latter option has as a prerequisite that the CoVs and Flu viruses circulating in animals also possess (probably acylated) cysteines in the cytoplasmic tails of their membrane proteins and that newly emerged viruses use the same set of DHHC proteins in animals and humans. Therefore, we analyzed here whether the described acylation sites in the proteins of pathogenic human viruses are present in other members of the various Influenza and coronaviruses, especially in viruses with a zoonotic potential. We then asked whether the DHHC genes required for acylation of Flu A and CoV proteins are also encoded in the genomes of the animal reservoirs of Influenza viruses (water birds) and coronaviruses (bats) as well as in intermediate hosts of these viruses. We then compared the primary structure of the relevant DHHCs from animals with their human orthologs and located amino acid substitutions in the resolved or modelled 3D structure of these DHHCs. 

## 2. Results 

### 2.1. Analysis of S-Acylation Sites in HA and M2 of Human and Animal Influenza Viruses

Flu A viruses are divided into subtypes on the basis of the HA and neuraminidase (NA) serotype present on its surface. Water birds contain viruses belonging to all HA subtypes, only H1, H2 and H3 subtypes were transmitted to humans, whereas pigs contain H1 and H3 subtypes. HAs belonging to all HA subtypes (except avian H8 and H15 and bat H17 and H18, which were not investigated) are stoichiometrically S-acylated at three or four cysteines in the C-terminal region [34,69]. An analysis of all HA sequences, present in the NCBI database, revealed that all virus strains possess cysteines in this region and that the cysteine pattern, i.e., the exact localization of the cysteines in the cytoplasmic tail is completely conserved within each HA subtype [40]. We created a phylogenetic tree of the HA subtypes and indicated the cysteine pattern for each HA subtype in the tree (Figure 2). The predominant scheme is the presence of one cysteine at the end of the TMR and two at the end in the cytoplasmic tail (see Table 1 for two HAs and Appendix A for a consensus sequence at the C-terminus of all HA subtypes). This pattern (called CxCC) occurs in the HA subtypes, which have been identified in animals, including humans (H1, H2, H3, H7), but also in viruses which circulate only in birds (H5, H10, H14, H15). In addition, the two viruses recently identified in bats that define their own subtypes H17 and H18 also contain this cysteine arrangement [8,9]. Thus, the cysteine pattern is a trait of the virus lineage and does not depend on the host of the virus. Since HA subtypes with the predominant cysteine pattern CxCC belong to both of the main phylogenetic groups 1 and 2, one can speculate that it is the ancestral arrangement present in the common ancestor of all Flu A viruses, which has been estimated to exist ~1000 years ago [70]. This pattern is preserved in all subtypes belonging to group 2 HAs and in the clade of group 1 HAs that is formed by HA subtypes H1, H2, H5, H6, H17 and H18. The remaining HA subtypes, which occur exclusively in water birds, exhibit a slightly different pattern. H11 subtype HA acquired a fourth cysteine (denoted CCCC) and, thus, carries four acyl chains. In the clade formed by HA subtypes H13 and H16 the first cysteine was apparently lost creating the scheme XCCC. H13 and H16 carry the middle cysteine at this position, which is thus shifted two positions towards the N-terminus of HA (denoted CCXC).

Even the glycoproteins of the other members of the Orthomyxoviridae are acylated. HA of Flu B contain the two C-terminal cysteine residue (XXCC). HEF of the human virus Flu C one and HEF of Flu D, a virus that circulates in cattle but occasionally spreads to pigs [71], has two cysteines at the end of the TMR, but the cytoplasmic tails of HEF are shorter (Table 1 and Appendix A). In addition, two hitherto unclassified Influenza viruses, that were recently identified in fish and amphibia [72], contain two and three cysteine residues, respectively, in the cytoplasmic tail of HA. Both HAs possess the C-terminal cysteines including two highly conserved adjacent hydrophobic residues (Appendix A) which are essential for virus replication of a human Flu A virus in cell culture [40].

In summary, S-acylation is a highly conserved and an ancient trait of HA acquired early during evolution. The pattern of acylated cysteine residues is independent of the virus host, but determined by the HA subtype and, thus, apparently does not change if viruses cross the species barrier. 

In contrast to HA, a cysteine at position 50 of M2 is not present in ~14% of all M2 variants deposited in the NCBI database [54], but whether the absence of the cysteine might be a feature of the viral host has not been examined. One could imagine that palmitoylation of M2 might be essential for virus replication in certain hosts, but dispensable in others. To investigate this, we searched the NCBI Influenza virus resource database (https://www.ncbi.nlm.nih.gov/genomes/FLU/Database/nph-select.cgi#mainform) (accessed on 24 March 2021) for full-length M2 amino acid sequences, but restricting the search to specific hosts and HA and NA subtypes. 

Avian viruses belonging to HA subtypes H12, H14, H15 and H16 contain in every M2 variant a cysteine at position 50. Avian viruses belonging to the other HA subtypes exhibit substitutions at position 50 of M2, mostly to a phenylalanine or tyrosine, rarely to a serine. The frequency of cysteine exchanges varies between 8.5–12% in HA subtypes H5, H6, H8 and H9, but is lower (~2%) in avian viruses belonging to the remaining HA subtypes (Appendix A). 

Next, we analyzed M2 sequences from mammalian viruses. The three virus strains circulating in pigs (H1N1, H1N2, H3N2) contain mostly, but not exclusively (96–97%), a cysteine at position 50. A striking difference occurs in M2 of the two equine viruses:H7N7, which was first isolated in 1957, but is extinct since 1977 and H3N8, which emerged in 1963 and is still circulating today [73]. Every M2 sequence of the H7N7 subtype, but only 13% of the H3N8 subtype contain a Cys at position 50. It is always replaced by a Phe, but, interestingly, the first four equine H3N8 M2 sequences, which were collected between 1963 and 1972, contain a Cys at position 50. One might speculate that the first virus introduced into horses from an unknown source probably had a palmitoylated M2 protein, but it was subsequently lost since it is not required for viral fitness in horses. 

A virus from the equine H3N8 lineage was transmitted in 2003 to dogs and probably as a consequence most M2 sequences (43 from 45 (=87%), including the one from the first canine virus) contain a Phe at position 50. Only one canine virus recently identified in China (A/canine/Zhejiang/S34/2015(H3N8)) exhibits a Cys at position 50. Another lineage spreading between dogs since 2006 belongs to the H3N2 subtype [74]. All the 240 M2 protein sequences in the database contain exclusively a cysteine at position 50. Finally, the four available sequences of bat viruses contain a Phe at position 50 of M2.

We then looked for the frequency of cysteine 50 in M2 of human virus strains, which are all thought to be derived from avian viruses. When they emerged in the human population, they caused a pandemic, but subsequently they became the predominant strains responsible for the seasonal Influenza. Human H2N2 viruses emerged in 1957 causing the “Asian Flu” and were replaced in 1968 by the “Hongkong Flu” H3N2 strain, descendants of which are still circulating today. Both viruses contain exclusively (H2N2, 122 sequences) or predominantly (H3N2, 99.6% form 905 unique sequences) a cysteine at position 50 of M2, including the first human H3N2 virus in the database (A/Aichi/2/1968(H3N2). 

Influenza viruses belonging to the subtypes H1N1 caused three human pandemics: 1918 the “Spanish Flu”, 1977 the “Russian Flu” and 2009 the “swine Flu” [2,75]. The 1918 H1N1 virus and its descendants contain exclusively a cysteine (84 sequences) at position 50 and only one out of 454 unique M2 sequences (0.2%) of the pandemic “swine Flu” virus contain a serine instead of a cysteine. In contrast, M2 sequences of the “Russian Flu” H1N1 strains contain mainly (74%) a serine at position 50, the remainder being Cys. However, the first 100 viruses sampled between 1977 and 1993 including the first one (A/USSR/90/1977(H1N1) contain a cysteine at position 50. A virus with a Ser at position 50 appeared for the first time in 1995 (A/Beijing/262/1995(H1N1), 18 years after virus emergence.

Avian viruses, mostly highly pathogenic variants of the subtypes H5N5 and H7N9 occasionally infect poultry workers, but these viruses did not acquire the ability to be transmitted between humans [76]. The frequency of cysteine at position 50 is similar between human virus strains and the corresponding avian strains sequenced between 1996 and 2019 when these individual spill-over events occurred. Ten percent of human and 24% of avian viruses H5N1 viruses possess a cysteine at position 50 contain. The value is lower in H7N9 strains, 3.2% in avian and 2.8% in human viruses.

In summary, we found no strict correlation between the frequency of an acylated cysteine in M2 and the host and/or the HA subtype of the respective virus. The frequency of amino acids other than cysteine is usually low in all virus hosts and all HA subtypes. Only two strains possess mainly another amino acid at position 50, either a Phe in equine H3N8 viruses or a Ser in descendants of the “Russian Flu” H1N1 strain. However, the earliest viruses from these lineages contain a cysteine suggesting that the virus that started the epidemic had an acylated M2. The subsequent replacement of the cysteine indicates that acylation of M2 does not provide a strong fitness advantage. This is consistent with cell culture experiments, where virus mutants with deleted acylation site in M2 revealed small growth defects. However, acylation of M2 and HA synergistically affect virus release and virus mutants were attenuated in mice [52,53,54]. Of note, none of the viruses without a cysteine at position 50 possess a cysteine at a neighboring position. Thus, shifting of acylated cysteines, as it frequently occurs in HA, does not occur in M2.

### 2.2. Analysis of S-Acylation Sites in S and E of Human and Animal Coronaviruses 

We performed a similar analysis regarding the presence of S-acylation sites in the spike and E-protein of a selection of important CoVs, comprising all coronavirus genera and viruses isolated from different hosts (Appendix A for S and Appendix A for E). S of SARS-CoV-2, subgenus *sarbecovirus* of the β-CoVs, contains a cluster of ten cysteine residues at the beginning of the cytoplasmic tail, which is conserved in the two closely related viruses, RaTG13 from *Rhinopholus affinis* [23] and a virus from pangolins [24]. In S of SARS-CoV-1 as well as in the related SARS-like viruses from *Rhinopholus* bats, HKU3 [77], Rp3/2004 [78], 279/2005 [79], RsSHC014 and Rs3367 [11], one of the cysteines located in the middle of the cluster is substituted by an alanine. S of MERS, which belongs to the *Merbecovirus* subgenus of β-CoVs, contains seven cysteines in this region and this cysteine pattern is also present in the cytoplasmic tails of the closely related bat-associated viruses HKU4, HKU5 [80] 133/2005 [79] and NeoCoV [19], which were identified in the bats *Tylonycteris pachypus* and *Pipistrellus abramus, both* from the family *Vespertilionidae*. Members of the subgenus *Embecovirus* of β-CoVs, which encompass two human “common cold” viruses (HKU1 and OC43), a virus from cattle (Quebec) and the mouse hepatitis virus (MHV), contain eight or nine cysteine residues located at identical positions in the cytoplasmic tail. Likewise, the HKU9 virus of the subgenus *Nobecovirus* of β-CoVs, from *Rousettus leschenaulti* bats (family) *Pteropodidae* [80] contains eight cysteines, but at slightly different positions than members of the subgenus *Embecovirus*. 

Members of the genus α-CoVs contain eight to eleven cysteines in the cytoplasmic tails of S. This group encompasses the two other human “common cold” viruses 229E and NL63, a virus from dogs (Insavc-1), an important pathogen of cats (feline infectious peritonitis virus, FIPV), the virus CoV 512/2005 from the bat *Scotophilus kuhlii* (family Vespertilionidae) [79], which is highly similar to the porcine epidemic diarrhea virus as well as a the recently emerged Swine Acute Diarrhea Syndrome (SADS) virus, which jumped from bats to pigs [81].

γ- and δ-CoVs have their animal reservoir in wild birds, but occasionally spread to poultry and even to mammals [82]. Infectious bronchitis virus, a γ-CoV which is endemic in poultry farms contains seven cysteines in the cytoplasmic tail of S. Seven to eight cysteines are present in the spike of the γ-CoV porcine deltacoronavirus, an emerging enteropathogen of swine which probably originated from a sparrow CoV [83] and in S of HKU19 and HKU20, which have been identified in Wigeon, a dabbling duck and in Night herons, respectively [84]. Even the S-protein of an unclassified CoV recently identified in amphibia (Chinese water skink) contain seven cysteine residues [72].

Recently, the risk for spillover of viruses from animals into the human population was estimated in order to enable pandemic prevention and preparedness [85]. Among the top 20 viruses are the known zoonotic coronaviruses SARS-CoV-1 and SARS-CoV-2 as well as bat-associated CoVs, such as 229E, RP3 and HKU9, that were already mentioned. However, some newly detected α-and β-CoVs as well as hitherto unclassified viruses ranked within the top 50 of the list. The hosts of these potentially prepandemic viruses are either bats belonging to four different families or rodents from two families. All these viruses contain clusters of cysteine residues in the cytoplasmic tail of S (Appendix A). 

A similar picture emerges if the cytoplasmic tails of E proteins from these viruses are compared (Appendix A). All Sarbecoviruses contain three cysteines and most Merbecoviruses two, the localization in the cytoplasmic tail is conserved within the subgenera. Members of the δ-CoVs contain three to five cysteines in E and infectious bronchitis virus (IBV), a γ-CoVs two. The latter are located 13 residues away from the transmembrane region, but are nevertheless palmitoylated [86]. Likewise, the E-protein of the potentially prepandemic viruses also contain cysteine clusters (Appendix A).

In summary, both E and S of all the viruses analyzed here contain a cluster of cysteine residues at the beginning of the cytoplasmic tail. This cluster is present in viruses of all coronavirus genera, regardless of whether viruses are infecting humans, various mammals (including bats and rodents) or birds. This conservation of cysteine residues points to an important function, since S is otherwise highly variable due to antigenic drift and recombination. S even acquired the ability to use different cellular receptors [29]. The presence of the cysteine cluster in all coronavirus genera also suggests that S and E were already S-acylated in an ancestral coronavirus, which has been estimated at approximately 8100 BC by molecular clock analysis [84]. During diversification of CoVs into genera, subgenera and species, changes to the number and also the precise localization of cysteines in the cytoplasmic tail occurred. However, the cysteine pattern remains rather constant once viruses have diversified into (sub)genera. 

### 2.3. Number and Type of DHHC Genes in Animal Hosts of Influenza and Coronaviruses

Having shown that the palmitoylated cysteines in HA, S and E are present in all Influenza viruses and coronaviruses, regardless of their origin, we investigated whether the DHHCs required for their acylation in human cells are also encoded in the genomes of the animal reservoirs of these viruses. The reservoirs of influenza A viruses are water birds of the orders *Anseriformes* (ducks, geese, swan) and *Charadriiformes* (shorebirds, gulls, alcids). From here viruses occasionally spread to poultry, including chicken (*Gallus gallus*), turkey (*Meleagris gallopavo*) and quail (*Coturnix japonica*), which all belong to the order *Galliformes*. Infection of falcons (*Falco peregrinus*, order Falconiformes) and pigeons (Columbiformes) has also been described. Searching the NCBI database for annotated DHHC sequences of these avian species revealed that their genomes encode less DHHC genes than identified in humans, the number varies between 18 (turkey) and 20 in most other avian species (Table 2). Most avian genomes analyzed here lack the genes encoding DHHC11, 19 and 24; geese and the *Oxyura jamaicensis* duck lack DHHC5, and turkey lacks DHHC4 and 21. Pigs, another important reservoir of Flu A virus, encode 21 DHHC genes; DHHC4 and 11 are not present. The zig-zag eel and a toad, which harbor ancient, unclassified influenza viruses, encode 18 and 20 DHHCs, respectively; DHHC 11, 18, 19 are missing in the toad and DHHC1 and 7 is also missing in the eel.

From the various animal reservoirs and intermediate hosts of human CoVs, genomic sequences are available for horseshoe and pipistrelle bats, *Manis javanica* (pangolin) and camel. All these animals encode the same number of DHHC genes that are present in the human genome, except pipistrelle bat, which lacks DHHC11, which is also missing in the animal hosts of Flu A. 

In summary, all DHHCs required for acylation of HA and M2 (2, 8, 15 and 20) and S and E (8, 9, 20) in human cells are present in the genome of the reservoir hosts of Flu A and CoVs. Likewise, the gene of DHHC22 which is upregulated during an influenza virus [87] infection is also encoded in all hosts of Flu A. Interestingly, all the virus’ hosts lack a comparable set of DHHCs. All animals that have fewer DHHC genes than humans, lack the gene for DHHC11. Little is known about the function of human DHHC11, except that it might modulate the innate immune response to DNA viruses [88]. All animals (except pigs) which encode 21 or less DHHCs do not contain DHHC19. Human airway cell lines do not express DHHC19 [66] and, hence, it is irrelevant for palmitoylation of respiratory viruses. However, DHHC19 (together with DHHC2) is required for S-acylation of nsp1 of Chikungunya Virus, which is transmitted by mosquitoes and infects epithelial and endothelial cells in various mammals (including humans) and birds [89]. All animals (except the fish and amphibia species) which possess only 20 or less DHHCs lack the gene for DHHC24, but essentially nothing is known about the function of the protein in humans. 

### 2.4. Structural Comparison of Human DHHCs with the Ortholog Proteins in Animal Hosts 

Next, we aligned the amino acid sequence of human DHHCs with their orthologs from animals. From the various Flu A hosts, we chose pigs and chicken, farm animals of economic importance and Mallard duck and sandpiper since they carry a large number of different avian Flu A viruses [90,91]. In addition, chicken, Mallard duck and sandpiper belong to different families. The percentage of identical amino acids are depicted in Table 3; Table 4 and amino acid substitutions relative to human DHHCs are highlighted in the alignment (Appendix A). Salient features within the primary structure of human DHHCs, such as transmembrane regions, cysteine-rich regions (CRD), DHHC and TTXE motifs and the amino acids that contact the acyl chain inserted into the hydrophobic cavity are also highlighted in the sequence. To estimate whether a certain amino acid substitution might have functional consequences we used the known 3D-structure of DHHC15 and 20 [59] to locate non-conservative amino acid replacements in animal DHHCs. The structure of the other DHHCs were determined using the SWISS model (https://swissmodel.expasy.org/) in March 2021 using the structure of DHHC20 as the template, as described in Materials and Methods. 

#### 2.4.1. Structural Comparison of Human DHHCs with the Ortholog Proteins in Animal Hosts of Flu A

DHHC2 from pigs is almost identical (96.7%) in sequence to human DHHC2. One deletion and one substitution occur in the short N-terminal part and six, mostly conservative substitutions in the C-terminal part, which are both not resolved in the 3D structure (Appendix A). The core domain of porcine DHHC2 contains just one substitution (M57I), which is located in TMR2 (labeled as magenta stick in Figure 3A). Since the amino acid side chain at position 57 is not pointing towards the interior of the cavity, it is unlikely to be involved in fatty acid binding. DHHC2 from two wild birds and from chicken is only about 84% identical in sequence to human DHHC2, reflecting the larger phylogenetic distance between birds and humans compared to pigs and humans. However, most of the amino acid substitutions are also located in the C- and N-terminal part of DHHC2—the latter contains insertions of up to 23 amino acids in avian enzymes. Transmembrane region 1 and 2 contain 15 amino acid substitutions compared to human DHHC2. However, none of them involve a residue that has been determined to contact the acyl chain inside the hydrophobic cavity [59], they are highlighted in green in the sequence alignment. The two residues that seal the hydrophobic cavity and are supposedly involved in determining the fatty acid specificity are highlighted in purple. Three non-conservative substitutions in all three avian species are located in TMR2, M57V, A64M and M65L, but their side chains are located outside of the hydrophobic cavity of DHHC2 (labeled as magenta sticks in Figure 3A). In addition, the CRD of avian DHHC2 contains three non-conservative substitutions, L94S in helix 2, G254R, located between TMR4 and the amphiphilic helix and M268L which is in the amphiphilic helix, but does not change its amphiphilic character. The distribution of amino acid substitutions in the structurally not resolved C-terminal region of DHHC2 is also interesting. Parts of this region are predicted to be unstructured (underlined in the amino acid alignment) and this region contains most of the amino acid substitutions. The remainder of the C-terminal domain (including the last 15 amino acids) is completely conserved between all five species. It is also noteworthy that the first residue of the catalytically important TTXE motif downstream of TMR4 is a Ser in all DHHC2 variants. 

DHHC15 from pigs is 98.2% identical to human DHHC15. It contains just two conservative substitutions in TMR1 and TMR2, but in amino acids that are not involved in fatty acid binding and one at the beginning of the CRD (Appendix A). Avian DHHC15 proteins are only 75% identical to human DHHC15. Their N-terminus lacks the first six amino acids and the C-terminus exhibits large stretches of exchanged amino acids. Four non-conservative substitutions are present near the zinc-binding region (Q81V, N84G, Q85K and M150I) and two in helix 2 (M109I, M113I, see Figure 3B). In addition, many amino acid substitutions are present in all four transmembrane regions. However, they are not the residues which that mediate fatty acid binding in DHHC15 [59] and in the 3D-structure their side chains are not exposed to the hydrophobic cavity. 

DHHC20 from pigs and birds are 91.3% and 73–77%, respectively, identical to human DHHC20. Both contain exchanges and deletions in loop1, that connects TMR1 and TMR2 (Appendix A). The residues are exposed to the lumen of the ER/Golgi and thus unlikely to have any functional importance. Otherwise, porcine DHHC20 is very similar to human DHHC20, only two non-conservative substitutions occur in the cysteine rich domain; A113G in helix 3 and S147I near one zinc-binding region (Figure 3C). In avian (but not porcine) DHHC20 threonine 240 of the TTXE motif is replaced by a serine (TTIE > STIE), but a similar motif (STEL) is present in human (and other) DHHC2 and, thus, probably has no functional consequences. Besides the two residues 113 and 147 that are also substituted in porcine DHHC20, the avian proteins contain additional non-conservative substitutions. M71K at the end of TMR2, K82N near the zinc-binding region, N88K in helix 2 and F98E, S99R and Q100P in helix 3 of the cysteine-rich domain.

DHHC8 from humans is 93.3% identical to porcine DHHC8, but only 63–66% identical to avian DHHC8 and, thus, seems to be the most diverse of the DHHCs investigated here. However, DHHC8 is different from the other DHHCs, which are closely related in a phylogenetic tree of all DHHCs, since it contains a long extension at its C-terminus, which is predicted to be disordered and to contain a PDZ-binding motif at its C-terminal end [58]. Indeed, the overwhelming majority of the amino acid exchanges are located in the C-terminal region (Appendix A). *Gallus gallus* and Anas platyrhynchos, but not *Calidris pugnas* contain an insertion of 15 amino acids adjacent to the amphiphilic helix. *Anas platyrhynchos* exhibits a deletion of 100 amino acids and Sus scrofa an N-terminal extension of 34 amino acids. Furthermore, large stretches of amino acids are different, especially between human and avian DHHC8 and other small insertions and deletions exist. However, the core domain is very similar between human and animal DHHC8. Porcine DHHC8 exhibits just one amino acid exchange relative to human DHHC8, which affects a putative palmitoylation site at position 237 (Cys > Tyr) in the amphiphilic helix [92,93]. Avian DHHC8 exhibit 15 amino acid substitutions, mostly conservative replacements in the transmembrane regions that are unlikely to affect the functionality.

#### 2.4.2. Structural Comparison of Human DHHCs with the Ortholog Proteins in Animal Hosts of CoVs

DHHC9 is highly conserved (96–99%) between all four species. The amino acid sequence of the core domain, defined as the region from TRM 1 to TRM4 plus the following amphiphilic helix, which essentially corresponds to the crystalized structure of DHHC20 is identical in camels, pangolin, pipistrellus and humans (Appendix A). DHHC9 from *R. sinicus* contains just a single substitution in this region, an exchange of A to T at position 115. The C-terminal part beginning from residue 325, which is a possibly unstructured proline-rich region is more diverse. In *Pipistrellus kuhlii*, a sequence of 19 amino acids is exchanged and nine amino acids are inserted at the C-terminus. DHHC9 from the other animals contain from four to nine amino acid substitutions in this region.

DHHC8 is also highly conserved (89% to 92%) between humans and the four animal hosts of CoV. The sequence alignment of DHHC8 revealed many substitutions, small insertions and deletions, but mainly throughout the long, presumably unstructured C-terminal domain in all animal hosts of CoVs, even a long insertion of 55 amino acids in *Manis Javanica* (Appendix A). DHHC8 from *Manis javanica* also contains an insertion of 62 amino acids between TMR1 and TMR2. In contrast, the core domain is well conserved exhibiting mostly conservative substitutions. An exception is a deletion of one and the substitution of four amino acids at positions 119–122 in DHHC8 from camels, i.e., SHCSV is converted to -PLQR (Figure 4A). Since His is involved in coordinating one zinc ion and Cys the other, it is unclear whether DHHC8 from camels is able to bind these critical ions. Studies have shown that mutations of the zinc coordinating residues change the structural integrity of DHHC3 [94]. However, they are not essential for catalytic activity, since Akr1, a DHHC protein in yeast, lacks five of the cysteines and one of the histidines that constitute the two zinc fingers [95] and two other yeast DHHCs tolerate exchange of many, but not all of the eight zinc-coordinating residues [96,97]. Another remarkable substitution is the substitution of the probably palmitoylated cysteine 237 in the amphiphilic helix [92,93] by a tyrosine in all DHHC8 proteins from the four animals.

DHHC20 sequences are more diverse throughout all species; the sequence identity compared to human DHHC20 varies from 86% (*Pipistrellus*) to 91.4% (camels). DHHC20 from all these animals contain an almost identical insertion of 12 consecutive amino acids into the loop connecting TMR 3 and 4 (Appendix A). Since this part of the protein is exposed to the lumen of the ER or Golgi, it is most likely not important for substrate binding. The animal DHHC20 proteins contain also substitutions in the four TMRs. Most of them do not change the hydrophobic character nor the length of the amino acid side chain. An exception is the exchange of Phe 61 by Ser (TMR1) in *Rhinopholus* and camel, and Ala by Thr at positions 187 (TMR3) and 218 (TMR4) in all species except pangolin, but the side chains of these residues are not directed towards the lumen of the hydrophobic cavity (Figure 4B). Thus, it seems unlikely that these substitutions affect fatty acid binding, also because these residues are not the ones that contact the acyl chain [59]. Two exchanges occur at the beginning and end of helix 3, Q100P in *Rhinopholus* and camel and A131D in all animals. Another exchange in the cysteine-rich domain of all animals (S147I) is located penultimate to a cysteine which is involved in zinc-binding. In addition, Cys 262 in the amphiphilic helix, a putative palmitoylation site [59,92,93] is exchanged in DHHC20s from all four species. There are also mostly conservative substitutions at the C-terminal part of DHHC20; *Pipistrellus kuhlii* contains a Pro-rich extension at the C-terminus of the protein.

In summary, the structural analysis revealed that most functional elements of human DHHCs are present in the orthologues from animals. An exception might be changes at the two zinc-binding sites in DHHC8 of camels, but it is unclear whether this affects coordinating the ions or the catalytic activity of the enzyme. DHHC20 of all three avian species exhibit a substitution in the essential TTXE motif, TTIE is changed to STIE. However, since Ser is also able to from a hydrogen bond with the DHHC motif of the enzyme, its catalytical activity is most likely not compromised. Accordingly, all the DHHC2 enzymes investigated here (including human DHHC2) also contain a Ser at the first position of the TTXE motif. Many DHHCs from animals (except DHHC8 and 9) contain substitutions in the four transmembrane regions compared to human DHHCs. However, since the affected residues are not exposed to the hydrophobic cavity, they are unlikely to affect fatty acid binding. In fact, the amino acids identified to contact the acyl chain in human DHHC15 and 20 [59] are conserved in all species. 

Which domains in DHHCs recognize and bind the protein substrates is still mysterious. The cysteine-rich domain located between TMR 2 and 3 might be involved, since it exhibits deletions and insertions if individual human DHHCs are compared, especially in the three α-helices preceding the zinc-binding domain [59]. However, indels in this region did not occur between the corresponding human and animal DHHCs. If animal DHHCs contain insertions, they are located between TMR 1 and 2 (avian DHHC20) or between TMR3 and 4 (bat DHHC20) which are unlikely to be involved in substrate recognition since they are exposed to the lumen of the ER or Golgi. 

The largest differences between human and animal DHHCs are located in the N- and C-terminal regions, which are not resolved in the crystal structures. These extensions often contain binding motifs to recruit cytosolic proteins to a DHHC protein to facilitate acylation [62,63]. However, the viral substrates are transmembrane proteins that are transported along the exocytic pathway and thus will encounter the DHHCs during the transit. The C-terminal regions of DHHCs 2, 8, 9 and 15 are predicted to be unstructured [58]. Such intrinsically unstructured regions, which adopt a broad range of transient conformations, are often involved in protein interactions and are able to bind to several different targets through an induced-fit mechanism [98]. It is unknown whether the unstructured regions of the DHHCs are involved in substrate binding and whether this is affected by amino acid substitutions. 

## 3. Discussion

Coronaviruses from all genera and from all virus hosts contain clusters of cysteine residues in the cytoplasmic tails of the main glycoprotein S and the ion channel E (Table 1 and Appendix A). Likewise, the HA of influenza viruses exhibits the same characteristic (Appendix A). This indicates that S-acylation of these cysteines is a very ancient feature acquired early during virus evolution which was fixed in the virus population since it increased viral fitness. This assumption is consistent with experimental data showing that S-acylation of the viral proteins is essential for virus replication in cell culture. The number of cysteines and their localization relative to the end of the transmembrane region differs slightly within each virus family, but remains constant within HA subtypes of Flu A (Figure 2, Appendix A) and partially between CoV genera.

Likewise, we reported recently that the envelope proteins GP5 and M of the Arterivirus porcine respiratory and reproductive syndrome virus (PRRSV) are acylated at three or two cysteines at the beginning of their cytoplasmic tails and that the modification is essential for virus budding [99]. Other members of the *Arteriviridae* isolated from various other species, such as equine arteritis virus, simian hemorrhagic fever virus and murine lactate dehydrogenase elevating virus also contain at least two cysteines in this region, but their precise localization varies between viruses.

In contrast, a cysteine as acylation site at position 50 is not present in around 14% of all M2 molecules. We did not find evidence that acylation of M2 is strictly required for replication in certain hosts. A human H1N1 lineage and the H3N8 equine strain contain predominantly another amino acid at position 50, but the virus that started the epidemic had a cysteine (Appendix A). The reason for the elimination of cysteine in M2 might not be evolutionary selective forces, but a “genetic bottleneck” effect. In a certain transmission event, an infected person or animal released droplets that contained only a limited virus population which does not represent the whole virus “swarm” replicating in his body. This droplet lacking a virus with an acylated M2 initiated an infection in another person or animal and created a new and different virus population by a “founder” effect. This suggests that acylation of M2 does not confer a strong selective fitness advantage for the virus. This is again consistent with cell culture experiments, which did not reveal a strong effect of removal of acylation sites for virus replication [100], suggesting that S-acylation does not provide a fitness advantage.

The DHHCs identified to acylate HA, M2, S and E in human cells are also present in the genome of all animal hosts of Flu A and CoVs. A sequence comparison of human with animal DHHCs and mapping the differences onto the 3D-structure of the DHHC enzyme did not reveal amino acid substitutions that would impact the functionality of the enzyme (Figure 2; Figure 3). It is, thus, tempting to speculate that the DHHCs identified to acylate HA, S and E in human cells also process these proteins in cells from other virus hosts. There is indirect evidence for that assumption for HA of Flu A viruses, since they are stoichiometrically acylated if purified from embryonated hen’s eggs as well as from mammalian cells. To take this idea further, we speculate that an ancient avian Flu A virus was already adapted to DHHC 2, 8, 15 and 20 (or some of these enzymes) before the lineage diversified into different subtypes and species and before some viruses spread to other animals and to humans. Amino acid exchanges occurring during virus diversification apparently has no effect, or a very slight effect, on substrate recognition by these DHHCs, since HA of the H1 and H7 subtypes which are acylated by the same set of DHHCs, differ in many amino acids surrounding the acylation sites (shaded grey in Table 1). An early adaption to a certain set of DHHCs during virus evolution might explain why DHHC2, 8, 15 and 20 are not required for acylation of HA of Flu B and HEF of Flu C [66]. The reservoirs of both viruses are humans where they might use other currently unidentified DHHC enzymes. 

Applying the same reasoning for the S-acylated proteins of coronaviruses, we speculate that an ancestral virus adapted to the same DHHCs that they are using in human cells, namely DHHC8, 9 and 20. Although α/β-CoVs and δ/γ-CoVs have now different animal reservoirs, bats and birds, respectively, a common ancestor of all CoVs existed around 10,000 years ago. However, it is not known whether the first CoV occurred in bats and jumped to birds or vice versa [84]. Note that both birds and bats contain the genes required for acylation of S and E in human cells. Alternatively, acquisition of acylation sites might be an example of convergent evolution, i.e., it occurred independently in different hosts and hence different sets of DHHCs might be involved. In any case, it seems likely that at least the α- and β-CoVs that pose the greatest risk to human health use the same DHHCs in bat and human cells. 

In summary, we propose that any of the Flu A and α- and β-CoVs circulating in their animal hosts require protein palmitoylation for replication and are using the same DHHCs in animal and human cells. This also suggests that DHHCs required for acylation of both S and HA, such as DHHC8 and 20, may be promising drug targets, since their inhibition probably blocks not only replication of the viruses already present in the human population, but also any other influenza and coronavirus, that might cross the species barrier in the future. 

However, no drug specific for a certain ZDHHC substrate interaction has been developed so far. 2-bromo-palmitate, which forms an irreversible covalent bond with the thiolate group of the catalytic cysteine in the DHHC motif (Figure 1) is widely used in research, but it also inhibits other enzymes of lipid metabolism [101]. By high throughput screening the compound V ([2-(2-hydroxy-5-nitro-benzylidene)-benzo[b]thiophen-3-one]) has been identified that inhibited the activity of four ZDHHC proteins for which biochemical assays were available. Since substance V inhibits also autoacylation of ZDHHC proteins, a step presumably essential for many ZDHHC-catalysed reactions, it is likely a general palmitoylation inhibitor [102,103].

A prerequisite for developing drugs specific for certain DHHC substrate pairs is an understanding of the three-dimensional structure of the contact site between the enzyme and the viral protein. As a first step in this direction, the amino acids in the viral protein that interact with the DHHC should be determined. The putative DHHC interacting surface of a viral spike glycoproteins is rather limited in size since it is composed of one (usually short) cytoplasmic tail and one transmembrane region. Having identified the DHHC interacting amino acids in a viral protein it might allow identifying the corresponding binding site in the DHHC protein. Advancements in Cryo-EM [104] might be helpful to determine the structure of transmembrane regions of viral spike proteins, eventually in complex with a DHHC. It will be especially interesting to explore whether the DHHCs involved in acylation of HA and S recognize their viral substrates by the same domain. Inhibitors can then be developed that fit into this contact site to prevent lipid attachment and viral replication. Mass spectrometry can then be used to determine the whole palmitoylome of a cell and how it is affected by drug treatment. An inhibitor with the least effect on acylation of essential cellular proteins and the most effect on palmitoylation (and hence presumably virus replication) would be the most promising drug. 

In summary, the development of drugs that target palmitoylation of viral proteins is still in its infancy and requires more basic research. However, since DHHC genes have been linked to a variety of human diseases, especially cancer and neurological disorders [105,106,107,108], exploring the mechanism how DHHC enzymes recognize their substrates is of general importance.

## 4. Material and Methods

### 4.1. Collection and Comparison of Viral Protein Sequences

Influenza A and B virus HA and influenza C and D HEF amino acid sequences were extracted from the influenza virus resource database (https://www.ncbi.nlm.nih.gov/genomes/FLU/Database/nph-select.cgi#mainform) and aligned as described in [40]. A consensus sequence of the C-terminus was created for each Flu A HA subtype, for Flu B HA and for Flu C HEF, which is shown Appendix A. The phylogenetic tree shown in Figure 2 was created with MEGA software using MUSCLE for alignment and the neighbor-joining tree method with HA proteins for each subtype as specified in Appendix A.

M2 of Influenza A virus: The NCBI Influenza virus resource database was searched for full-length M2 amino acid sequences (97 residues), but restricting the search to specific hosts, HA and occasionally NA subtypes. To separate M2 sequences present in H1N1 viruses causing a certain pandemic and for the descendants of this virus the collection date was restricted to 1918–1976 (“Spanish Flu”), 1977–2008 (“Russian Flu”) and 2010–2020 (“swine flu”), respectively. The function “collapse identical sequences” was used before retrieved sequences were aligned using the multiple alignment function of the database and the number of sequences not having a cysteine at residues at position 50 were counted. Results are depicted in Appendix A.

Coronaviruses spike and envelope proteins: Amino acid sequences of envelope and spike proteins of various coronaviruses were collected from both UniProt at https://www.uniprot.org/ and National Center for Biotechnology Information (NCBI) in March 2021 as indicated by their accession numbers. The C-terminal 60 residues representing the transmembrane and cytoplasmic regions were used for Appendix A.

### 4.2. Collection and Comparison of DHHC Sequences

The keywords “DHHC” and the name of various bird species known to harbor avian influenza viruses were used to search the NCBI database https://www.ncbi.nlm.nih.gov/ accessed on 19 March 2021. DHHC sequences present in the genome of chicken and pigs were retrieved in a similar manner. 

The DHHC isoforms sequences of human, *Rhinolophus sinicus*, *Pipistrellus kuhlii*, *Camelus dromedarius* and *Manis javanica* were extracted from the coding sequences of the latest genome assembly of each species at the ftp ensemble genomes database http://ftp.ensembl.org/pub/release-103/fasta/. which was accessed on 6 March 2021.

From the various isoforms of each human DHHC the longest was chosen, except for DHHC8 to include the C-terminal PDZ motif. From the animal DHHCs, the isoform with a comparable number of amino acids was selected. The animal DHHC sequences were aligned with their human orthologs using the biological sequence alignment editor Bioedit and the ClustalW multiple sequence alignment method [109]. The percent amino acid identity of each DHHC sequence to human DHHC was obtained using Basic Local Alignment Search Tool (BLAST) for proteins (blastp) of National Center for Biotechnology Information (NCBI) at http://blast.ncbi.nlm.nih.gov on 8 March 2021. Amino acid differences to human DHHCs are highlighted in the animal DHHCs in Appendix A. In the human sequence several features are highlighted according to the crystal structures of DHHC 15 and 20 [59].

### 4.3. Structural Analysis of Amino Acid Substitutions between Human and Animal DHHCs

Figures of DHHC proteins were created with PyMol (Molecular Graphics System, Version 2.0 Schrödinger, LLC, https://pymol.org/2/). The pdb-files 6BML (human DHHC20) and 6BMS (zebrafish DHHC15) were used to highlight differences in the DHHCs from animals in the 3D-structures based on the sequence alignments. The hydrophobic amino acid contacting the acyl chain are described in (36) and the amino acid sequence alignment in the same reference was used to find amino acids in equivalent positions in human DHHC2 and DHHC15. 

The model of DHHC8 was created with SWISS-model using the first 275 amino acids of human DHHC8 as target sequence. SWISS-model found the human DHHC20 structure (pdb file 6bmm.1. A) as the best template for modeling. The target sequence of DHHC8 and DHHC20 exhibit a sequence identity of 27.20% and a sequence similarity of 35%. Although the global QMEAN (Qualitative Model Energy Analysis) is quite low (−5.17) the local absolute quality estimates of the four transmembrane regions is much better (inset in Figure 4A). 

The model of DHHC2 was also created with Swiss-model using the structure of DHHC20 (pdb 6bml.1.A) as the template. The target sequence of DHHC2 and DHHC20 exhibit a higher sequence similarity of 53%, since DHHC2 and DHHC20 (and also DHHC15) are closely related in a phylogenetic tree of all DHHCs. As a consequence, the quality of the whole model is high (QMEAN: −1.81), and the structure of the transmembrane regions has been predicted with particularly high confidence (blue color in the inset of Figure 3A).

## Figures and Tables

**Figure 1 pathogens-10-00669-f001:**
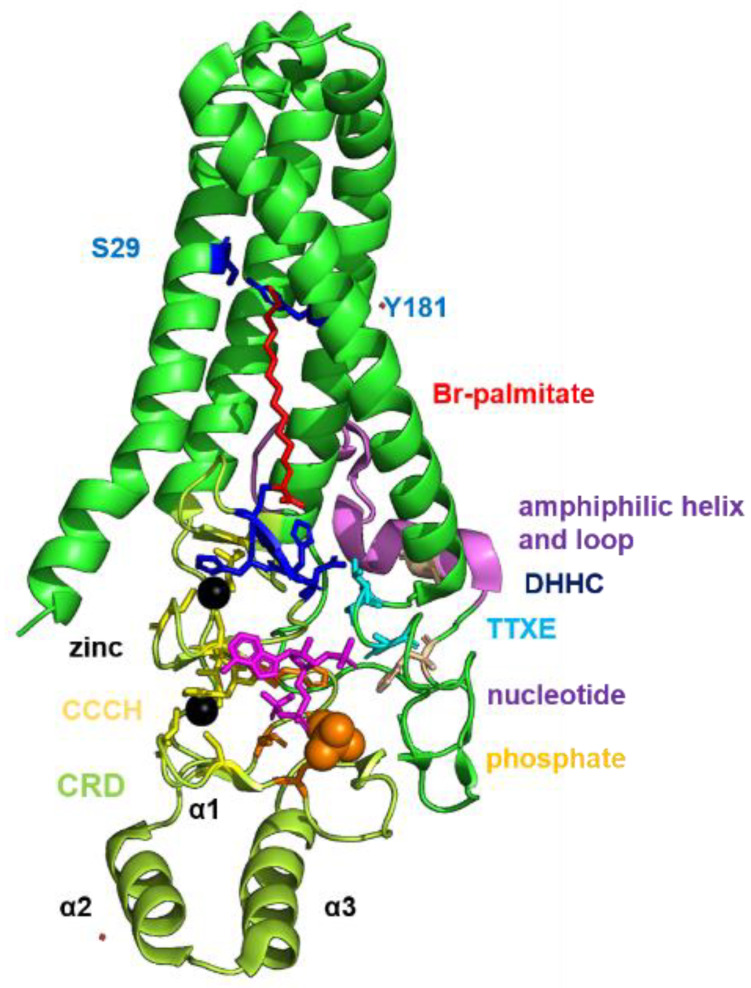
Crystal structure of human DHHC20. The location of relevant regions of DHHC20 are highlighted and labeled. CRD: cysteine-rich domain in light green. CCCH are cysteine and histidine residues (shown as yellow stick) that bind two zinc ions, shown as grey sphere. DHHC is the catalytically active motif (blue sticks), the Cys is covalently linked to bromo-palmitate (red) which represents an intermediate of the acylation reaction. The amino acids S29 and Y 181 (shown as blue sticks within the green transmembrane regions) seal the hydrophobic cavity and are thought to determine the fatty acid specificity. The TTXE motif is shown in light blue. A phosphate group and a nucleotide (more precisely 5′-diphosphoadenosine3′-phosphate) are non-covalently bound to the CRD, which are supposed to be the binding site of CoA of the acyl-CoA substrate.

**Figure 2 pathogens-10-00669-f002:**
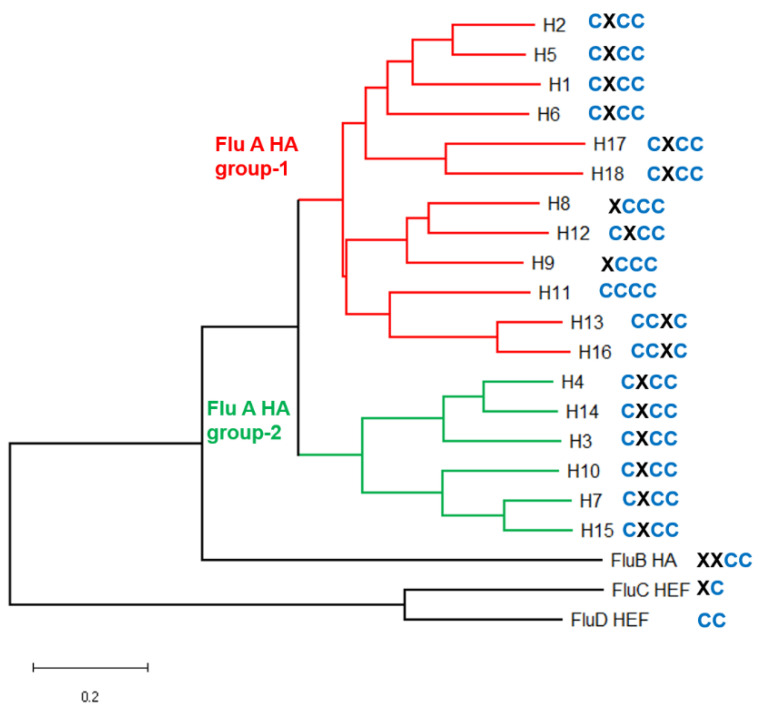
HA-based phylogenetic tree of Influenza viruses and location of acylated cysteine residues. The tree was created with representative HA sequences from all Flu A HA subtypes, from Influenza B virus and HEF sequences from Influenza C and D virus (see Appendix A for accession numbers). CXCC indicates the location of cysteine residues. The first residue is located at the end of the transmembrane region, the other three in the cytoplasmic tail. See Appendix A for a consensus sequence of the amino acids present in each Flu A HA subtype, in HA of Flu B and HEF of Flu C and D.

**Figure 3 pathogens-10-00669-f003:**
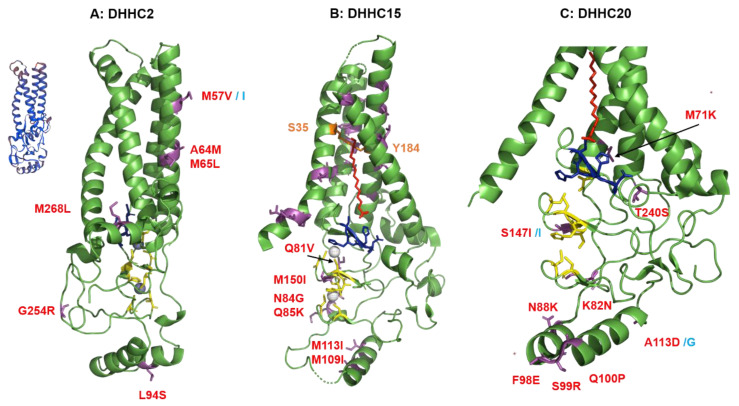
Amino acid substitutions between human DHHC and DHHCs in animal host of Flu A. (**A**) Computational model of human DHHC2. Non-conservative amino acids exchanges in porcine (labeled in blue) and avian DHHC2 (labelled in red) are highlighted as magenta sticks. DHHC2 is rotated by 180 degrees relative to the other structures in this figure showing the backside view of the molecule. The structure was created with Swiss-model (https://swissmodel.expasy.org/) on 23 March 2021 using the structure of DHHC20 (pdb 6bml.1.A) as the template. The quality of the whole model is quite high (GMQE: 0.62, QMEAN: −1.62), especially the structure of the transmembrane regions is predicted with high confidence (blue color in the inset). (**B**) Structure of DHHC15. The structure shows the model of zebrafish DHHC15 (pdb 6BMS) with Br-palmitate (red) inserted into the hydrophobic cavity covalently bound to Cys of the DHHC motif (shown as blue sticks). Cys and His residues that coordinate zinc ions (white spheres) are shown as yellow sticks. Non-conservative amino acids exchanges in avian DHHC2 are highlighted as magenta sticks. (**C**) Structure of DHHC20. The structure shows the model of human DHHC20 (pdb 6BML) with Br-palmitate (red) inserted into the hydrophobic cavity covalently bound to cys of the DHHC motif (blue sticks). Cys and His residues that coordinate zinc ions (white spheres) are shown as yellow sticks. Non-conservative amino acids exchanges in porcine (labeled in blue) and avian DHHC2 (labelled in red) are highlighted as magenta sticks.

**Figure 4 pathogens-10-00669-f004:**
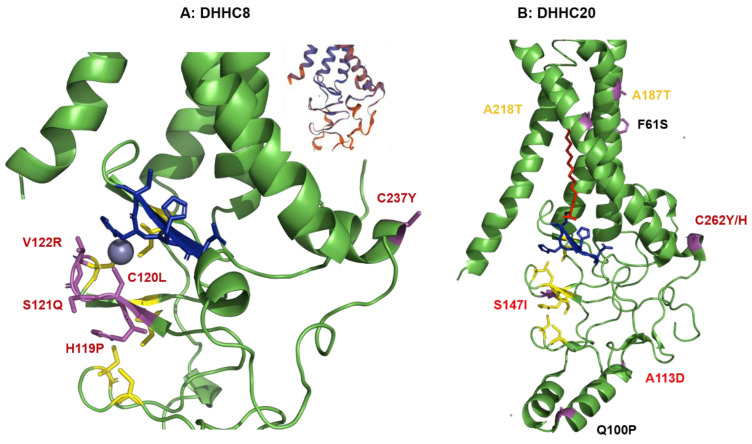
Amino acid substitutions between human DHHC and DHHCs in animal host of CoVs. (**A**) Computational model of human DHHC8. Non-conservative amino acids exchanges in DHHC2 from are highlighted as magenta sticks. The model of ZDHHC8 was created with SWISS-model using the first 275 amino acids of human DHHC8 as target sequence using the human DHHC20 structure (pdb file 6bmm.1. A) as the template. The global QMEAN, (Qualitative Model Energy Analysis) of the whole model is quite low (−5.17), but locally the quality differs as shown by the inset, where blue indicates higher and red lower quality. (**B**) Structure of DHHC20. The structure shows the model of human DHHC20 (pdb 6BML) with Br-palmitate (red) inserted into the hydrophobic cavity covalently bound to Cys of the DHHC motif (blue sticks). Cys and His residues that coordinate zinc ions (white spheres) are shown as yellow sticks. Non-conservative amino acids exchanges in animal DHHC20 are highlighted as magenta sticks. The residue and the substitutions are labeled in red, if the residue is exchanges in all four species; in orange if it is exchanged in all species except pangolin and in black if exchanged in Rhinolophus and camel.

**Table 1 pathogens-10-00669-t001:** Acylation sites in structural proteins of influenza and coronaviruses. The transmembrane region is underlined. Acylated cysteine are highlighted in yellow. Highlighted in grey are amino acid differences between the HAs of two different virus stains, which are acylated by the same DHHC proteins. Flu A HA/H7 is the sequence of the strain A/Fowl plague virus/Rostock/8/1934 H7N1; UniProtKB-P03459 and Flu A HA/H1 from the human strain A/Wilson-Smith/1933 H1N1, UniProtKB-P03454 and the M2 sequence is from the same strain, UniProtKB—P05780. See Appendix A for more sequences and the other accession numbers.

**S-Acylation Sites in Spike Proteins of Influenza and Coronaviruses**		
Flu A	HA/1	LVSLGAISFWMC SNGSLQCRICI
Flu A	HA/7	LAIAVGLVFICV KNGNMRCTICI
Flu B	HA	LMIAIFIVYMVS RDNVSCSICL
Flu C	HEF	LAALVISGIAIC RTK
MHV	S	AGVAVCVLLFFI CCCTGCGSCCFKKCGNCCDEYGGHQDSIVIHNISSHED
MERS-CoV	S	LVALALCVFFIL CCTGCGTNCMGKLKCNRCCDRYEEYDLEPHKVHVH
SARS-CoV-1	S	GLIAIVMVTILL CCMTSCCSCLKGACSCGSCCKFDEDDSEPVLKGVKLHYT
SARS-CoV-2	S	GLIAIVMVTIML CCMTSCCSCLKGCCSCGSCCKFDEDDSEPVLKGVKLHYT
**S-Acylation Sites in Ion Channels of Influenza and Coronaviruses**		
Flu A	M2	IIGILHLILWIL DRLFFKCIYRRFKYGLK
MHV	E	VTIIVVAFLASI KLCIQLCGLCNTLVLSPSIYLYD
MERS-CoV	E	TLLVCMAFLTAT RLCVQCMTGFNTLLVQPALYLYN
SARS-CoV	E	FLLVTLAILTAL RLCAYCCNIVNVSLVKPTVY
SARS-CoV-2	E	FLLVTLAILTAL RLCAYCCNIVNVSLVKPSFYVYS

**Table 2 pathogens-10-00669-t002:** Number and type of DHHC genes in the genome of hosts of Flu A and CoVs.

Order or Family	Species	# DHHCs	DHHC Missing	Host of
Primate	Humans (*Homo sapiens*)	23	X	Flu A
Suidae	Pigs (*Sus scrofa*)	21	4, 11	Flu A
Anseriformes	Mallard (*Anas platyrhynchos*)	20	11, 19, 24	Flu A
Anseriformes	Duck (*Aythya fuligula*)	20	11, 19, 24	Flu A
Anseriformes	Duck (*Oxyura jamaicensis*)	19	5, 11, 19, 24	Flu A
Anseriformes	Gesse (*Anser*)	19	5, 11, 19, 24	Flu A
Anseriformes	Swans (*Cygnus*)	20	11, 19, 24	Flu A
Charadriiformes	Sandpiper (*Calidris pugnax*)	20	11, 19, 24	Flu A
Galliformes	Chicken (*Gallus gallus*)	20	11, 19, 24	Flu A
Galliformes	Turkey (*Meleagris gallopavo*)	18	4, 11, 19,21, 24	Flu A
Galliformes	Quail (*Coturnix japonica*)	20	11, 19, 24	Flu A
Falconiformes	Falcon (*Falco peregrinus*)	20	11, 19, 24	Flu A
Columbiformes	Pigeon (*Columba livia*)	20	11, 19, 24	Flu A
Mastacembelidae	Zig-zag eel (*M. armatus*)	18	1, 7, 11, 18, 19	Flu
Bufonidae	Toad (*B. bufo*)	20	11, 18, 19	Flu
Manidae	Pangolin ( *Manis javanica* )	23	X	SARS-2
Camelidae	Camel (*Camelus dromedarius*)	23	X	MERS
Rhinolophidae	Horseshoe bat (*Rhinolophus s*.)	23	X	SARS-1 +2
Vespertilionidae	Pipistrelle bat (*P. kuhlii*)	22	11	MERS

The table lists the number (#) of DHHCs present in the genome of hosts of Flu A and CoVs, the name of the species and the family or order to which it belongs. The human DHHC not present in the respective animal species are also listed.

**Table 3 pathogens-10-00669-t003:** Percent amino acid identity of DHHCs in Flu A hosts compared to human DHHCs.

	Pigs (Sus scrofa)	Chicken (Gallus gallus)	Sandpiper (Calidris pugnax)	Mallard Duck (Anas platyrhynchos)
**DHHC2**	96.7%	84.2%	84.5%	84.5%
**DHHC8**	93.3%	65.1%	62.8%	66.2%
**DHHC15**	98.2%	75.3%	75.9%	75.3%
**DHHC20**	91.3%	77.7%	76.6%	73.1%

Percent amino acid identity of the DHHC involved in acylation of HA and M2 of Flu A in four important hosts of the virus compared to the human DHHC.

**Table 4 pathogens-10-00669-t004:** Percent amino acid identity of DHHCs in CoV hosts compared to human DHHCs.

	Horseshoe Bat (Rhinolophus sinicus)	Pangolin (Manis javanica)	Camel (C. dromedarius)	Pipistrelle Bat (P. kuhlii)
**DHHC8**	89.31%	90.00%	92.38%	88.82%
**DHHC9**	96.98%	98.63%	98.08%	95.99%
**DHHC20**	87.27%	88.54%	91.39%	86.17%

Percent amino acid identity of the DHHC involved in acylation of S and E of SARS-CoV-2 in three important hosts of SARS-CoV-1, SARS-CoV-2 and MERS compared to the human DHHC.

## Data Availability

All data are available in the manuscript including the Appendix A.

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
