# Peer review of "S-Acylation of Proteins of Coronavirus and Influenza Virus: Conservation of Acylation Sites in Animal Viruses and DHHC Acyltransferases in Their Animal Reservoirs"

_pathogens, 2021, doi:10.3390/pathogens10060669_

Round 1

Reviewer 1 Report

This manuscript on S-acylation of proteins from Coronaviruses and Flu is very thorough and well written.

This work provides a very nice framework for future work to focus on the significance of acylation and viral replication and it's importance in the field. And it of course suggests that the DHHCs may be a good target to block these viruses and this would help guide future research in this area. 

What I think is sorely missing and should be considered to be added is something about DHHC inhibitors. Questions that come to mind are 1) are there any? 2) If not why not...too difficult to make 3) If yes then what are they? 4) Are they general or specific? 5) If general inhibitors can we make them more specific? 6) how does your research in this manuscript address these points?

At one point they say Manuscript accepted (this should have some sort of reference at this point)

As far as editing of the writing I only noted an "s" on the "leaves" (line 35) and "s" on the word jumps (line 36). it would read more appropriately with the "s"'s removed.

On line 127 they mention work of several groups and so at the end of this sentence they should put the refs they are referring to there.

Line 287 has an extra space or two.

Line 391 they should also add the Plos Pathogens paper on the concern for CoVs infecting bats of North America

Author Response

Reviewer 1:

This manuscript on S-acylation of proteins from Coronaviruses and Flu is very thorough and well written.

This work provides a very nice framework for future work to focus on the significance of acylation and viral replication and it's importance in the field. And it of course suggests that the DHHCs may be a good target to block these viruses and this would help guide future research in this area. 

What I think is sorely missing and should be considered to be added is something about DHHC inhibitors. Questions that come to mind are 1) are there any? 2) If not why not...too difficult to make 3) If yes then what are they? 4) Are they general or specific? 5) If general inhibitors can we make them more specific? 6) how does your research in this manuscript address these points?

We added an additional paragraph shortly describing a strategy how to develop DHHC inhibitors that are specific for one or several closely related DHHCs to the end of the discussion.

At one point they say Manuscript accepted (this should have some sort of reference at this point).

The manuscript is now published and the reference is added.

As far as editing of the writing I only noted an "s" on the "leaves" (line 35) and "s" on the word jumps (line 36). it would read more appropriately with the "s"'s removed.

Corrected.

On line 127 they mention work of several groups and so at the end of this sentence they should put the refs they are referring to there.

These are the references 37-40 which are cited two sentences later after description of the results.

Line 287 has an extra space or two.

Extra spaces were deleted.

Line 391 they should also add the Plos Pathogens paper on the concern for CoVs infecting bats of North America.

We added the reference to the manuscript, but to the introduction, line 74.

“There might be the risk for reverse zoonotic transmission of SARS-CoV-2 to bats in North America where they might establish a new wildlife reservoir [25].”  

Reviewer 2 Report

One major issue of proposing DHHCs as targets for pan-antiviral drug development is the large number of DHHC family.  In human 23 of DHHCs are present, and SARS-CoV-2 and Flu A proteins are acylated by an overlapping set of DHHCs. Do the authors proposed to block all of them or only selective members?  

  1. Are DHHC SNPs correlated with Flu/Cov viral infections/disease severity?
  2. line 215-217, the authors mentioned that DHHCs required for acylation of HA could be promising drug targets, since their blockade should result in suppression of viral replication, while acylation of cellular proteins will not be (or very little) compromised. Please elaborate how that could be achieved.
  3. line 342-349, the following observation does not support the rationale of this study: no correlation of the frequency of an acylated cysteine with the host and /or HA-subtype of the virus, which is not consistent with line 660 “The number of cysteines and their localization relative to the end of the transmembrane region differs slightly within each virus family, but remains constant within HA-subtypes of Flu A”
  4. line 634, “Which domains in DHHCs recognize and bind the protein substrates is still mysterious.” This will pose a big challenge for DHHCs as drug targets.
  5. Acylation of M2 (but not HA) does not provide a strong fitness in flu. Do you have a mechanism/hypothesis  for this?
  6. line 87, reference on structure of SARS-CoV-2 spike protein should be cited.

Wrapp D   et al. “Cryo-EM structure of the 2019-nCoV spike in the prefusion conformation” Science, 2020 Mar 13;367(6483):1260-1263

  1.  line 202-203, reference should be list in the reference list at the end.

Author Response

Reviewer 2

One major issue of proposing DHHCs as targets for pan-antiviral drug development is the large number of DHHC family.  In human 23 of DHHCs are present, and SARS-CoV-2 and Flu A proteins are acylated by an overlapping set of DHHCs. Do the authors proposed to block all of them or only selective members? 

We added an additional paragraph shortly describing a strategy how to develop DHHC inhibitors that are specific for one (or a few) DHHC viral substrate interactions to the end of the discussion.

Are DHHC SNPs correlated with Flu/Cov viral infections/disease severity?

This is actually an interesting question, but we did not find any published paper where this topic was investigated.

line 215-217, the authors mentioned that DHHCs required for acylation of HA could be promising drug targets, since their blockade should result in suppression of viral replication, while acylation of cellular proteins will not be (or very little) compromised. Please elaborate how that could be achieved.

We added an additional paragraph shortly describing a strategy how to develop DHHC inhibitors that are specific for one (or a few) DHHC viral substrate interaction to the end of the discussion.

line 342-349, the following observation does not support the rationale of this study: no correlation of the frequency of an acylated cysteine with the host and /or HA-subtype of the virus, which is not consistent with line 660 “The number of cysteines and their localization relative to the end of the transmembrane region differs slightly within each virus family, but remains constant within HA-subtypes of Flu A”

Line 342-349 summarizes the investigation on the cysteine residues in M2, whereas line 660 the results on the acylation sites in HA. To make this clearer we added the following to the sentence in line 342-343.

In summary, we found no strict correlation between the frequency of an acylated cysteine in M2 and the host and/or the HA-subtype of the respective virus.

line 634, “Which domains in DHHCs recognize and bind the protein substrates is still mysterious.” This will pose a big challenge for DHHCs as drug targets.

Yes, it requires more basic research as outlined in the additional paragraph at the end of the discussion.

Acylation of M2 (but not HA) does not provide a strong fitness in flu. Do you have a mechanism/hypothesis for this?

We added the following speculation to the text:

“The reason for the elimination of cysteine in M2 might not be evolutionary selective forces, but a “genetic bottleneck” effect. In a certain transmission event an infected per-son or animal released droplets that contained only a limited virus population which does not represent the whole virus “swarm” replicating in his body. This droplet lacking a virus with an acylated M2 initiated an infection in another person or animal and created a new and different virus population by a “founder” effect.”

line 87, reference on structure of SARS-CoV-2 spike protein should be cited.

Wrapp D   et al. “Cryo-EM structure of the 2019-nCoV spike in the prefusion conformation” Science, 2020 Mar 13;367(6483):1260-1263

The reference was added.

 line 202-203, reference should be list in the reference list at the end.

The reference, a preprint, was added.

Round 2

Reviewer 2 Report

My concerns have been addressed.